# The Influence of COVID-19 on Community Disaster Resilience

**DOI:** 10.3390/ijerph18010088

**Published:** 2020-12-24

**Authors:** Wenping Xu, Lingli Xiang, David Proverbs, Shu Xiong

**Affiliations:** 1Evergrande School of Management, Wuhan University of Science and Technology, Wuhan 430065, China; xianglingli1017@163.com (L.X.); xs434991329@163.com (S.X.); 2Faculty of Computing, Engineering and the Built Environment, Birmingham City University, Birmingham B4 7BD, UK; David.Proverbs@bcu.ac.uk

**Keywords:** community disaster resilience, ISM-AHP model, key factors, COVID-19

## Abstract

Global pandemics, such as the Coronavirus Disease 2019 (COVID-19), have serious harmful effects on people′s physical health and mental well-being. It is imperative therefore that we seek to understand community resilience and identify ways to enhance this, especially within our cities and communities. Therefore, great emphasis is now placed on how cities prepare for and recover from such disasters, and community resilience has emerged as a key consideration. Drawing upon research on the theory of resilience, this study seeks to identify the factors that influence community resilience and to analyze their causation toward helping to manage the risks associated with the COVID-19 pandemic. Seventeen factors from the five dimensions of social capital, economic capital, physical environment, demographic characteristics, and institutional factors are used to construct an index system. This is used to establish the structural level and importance of each factor. Data were collected using a questionnaire survey involving 12,000 members of key community groups in the city of Wuhan. An interpretative structural model (ISM) combining the analytic hierarchy process (AHP) method was then used to obtain the multi-level hierarchical structure composed of direct factors, indirect factors, and fundamental factors. The results show that the income level, vulnerability of the population, and the built environment are the main factors that affect the resilience of communities affected by COVID-19. These findings provide useful guidance toward the effective planning and design of urban construction and infrastructure. The results are expected to be useful to inform future decision-making and toward the long term, sustainable management of the risks posed by COVID-19.

## 1. Introduction

In recent years, earthquakes, floods, droughts, public health epidemics, and other emergencies have become frequent occurrences [1]. These events can seriously affect the normal life of community residents and directly or indirectly cause many casualties and huge economic losses. Public health outbreaks of the Coronavirus Disease 2019 (COVID-19) around the world in early 2020 have posed a serious threat to the world′s population, and so far, remain uncontrolled in many regions. According to the COVID-19 situation dashboard of the World Health Organization (WHO), the virus has spread to 216 countries, infected more than 28,800,000 people worldwide to date, and has also killed over 900,000 [2]. Simultaneously, it has brought a series of social, economic, and environmental impacts to countries around the world [3].

In this pandemic, strict restrictions and policies to prevent people meeting in groups have been adopted in China. Nevertheless, there are still many problems and challenges in the prevention and control of the COVID-19 epidemic [4]. It is therefore necessary to develop approaches to systematically learn from the experience and improve our resilience to future pandemics of this nature. This demands a rigorous and robust process by which we can evaluate the experience, and our responses and actions, in order that we can develop improved prevention and control measures for similar global health incidents in the future. This is in line with the need to improve the adaptability, absorbability, resistance, and recovery of communities, which has become a contemporary topic in the field of city planning, emergency management, and disaster management, or what has broadly been termed “urban resilience,” or specifically “community disaster resilience” [5,6].

Resilience is a fairly modern concept in the context of disaster management and recognizes the need to mitigate the damage and duration of disasters to support the functionality of the emergency management systems [7]. This also acknowledges the importance of the community as the first location and basic defense line of the disaster response. Thus, the resilience of communities and their response to disasters has become an important component of disaster prevention and mitigation. 

In 1973, the theory of resilience was first applied to the field of ecology by Holling [8] and has been enriched by scholars in the disciplines of social-economics [9], social-ecology [10], engineering [11], geography [12,13], and most recently, urban planning [14,15]. Related research has included climate change, terrorist attacks, epidemic diseases, resource depletion, and other aspects. Community disaster resilience is one of the mainstream research subjects of disaster resilience [16] and is defined as a concept that “enhances the ability of a community to prepare, absorb, recover, and more successfully adapt to actual or potential adverse events in a timely and efficient manner” [7]. In fact, this definition implies the need to deal with uncertain risks by applying hard engineering measures, but its prerequisites should first be to fully evaluate and analyze the disaster resilience of the community. Conceptual frameworks of disaster resilience are abundant, and these are constantly being innovated and developed.

The Baseline Resilience Indicators for Communities (BRIC) model proposed by Cutter et al. [17] is widely applied and provides a series of secondary indicators to evaluate community resilience. BRIC is a quantifiable index and has been applied in several different regions, such as Taiwan, Norway, and Australia [17,18,19,20]. The model has mainly been used to identify appropriate strategies for building and enhancing community resilience. Based on this model, the differences of disaster resilience between urban and rural places were concluded [21]. Bruneau [22] established an evaluation framework of community resilience, which mainly concerned evaluating resilience characterized by area; Soetanto [23] proposed an assessment method of community resilience, a key step to reduce disaster risk and better respond to natural hazards, by developing community resilience assessment tools; Scott [24] proposed the construction of a simulation model of community disaster resilience through the participation of relevant stakeholders, which was combined with discrete event simulation and a new visual analysis interface to predict system changes through simulation modeling, scenario development, and the intervention of experts and stakeholders.

Compared with other countries, research on community resilience in China has only recently been undertaken with somewhat limited application of the theories developed in parts of Europe and the U.S. For example, Ouyang et al. [25] divided resilience into three stages to evaluate urban resilience. Xu et al. [26] studied the resilience of communities to urban flooding and put forward a comprehensive elastic index model. Wang et al. [27] defined the concept of community resilience and proposed a means for its measurement. Zhang et al. [28] put forward countermeasures and suggestions on the construction of an Urban Flexible community in China based on the interpretation of the theory of flexible community. Li et al. [29] studied an evaluation index system of disaster resilience of a community composite ecosystem. Meng et al. [30] conducted an elastic evaluation of an existing community in the city of Tianjin based on the data elastic evaluation system and found that the communities generally have community management problems, such as low awareness of disaster prevention, incomplete space construction, and inflexible emergency management; Liao et al. [31] learned from international examples of community development, from the policy level to the national level. They proposed recommendations for the development of community resilience in China based on five aspects: building open community space systems, coordinating community governance, building both a “smart community” and a “sponge community,” and improving a community’s self-organization ability. Li et al. [32] developed an index system based on the Baseline Resilience Indicators for Communities (BRIC) and the judgment of experts and evaluated this across 288 cities. Collectively, these research findings have promoted the understanding of community resilience theory in Chinese academia. However, there remains a lack of theoretical and practical research on community resilience especially in the context of pandemics such as COVID-19. Research in the fields of health and disease and environmental epigenetics has revealed that adverse social and material conditions during early life can increase the later risk of noncommunicable diseases (NCDs) [33]. These conditions include socioeconomic inequalities and physical conditions associated with infrastructure and construction. Therefore, social justice is fundamental to promoting health in society. Greater resilience to health emergencies requires a systemic understanding of these matters in order to support the ability to “bounce back” from these events.

In the process of constructing an evaluation model of urban community resilience, the logical relationship between the influencing factors should first be fully considered. Here, an evaluation model of urban community resilience index based on the interpretative structural model–analytic hierarchy process (ISM-AHP) method was first established. This is an interpretative structural model (ISM) that combines the analytic hierarchy process (AHP) method. The community is divided into three parts: Wuchang, Hankou, and Hanyang in the city of Wuhan. A typical community group was selected from each of these three parts of the community, and relevant data were obtained. Based on the actual characteristics of these three parts of urban communities, a ranking of the factors affecting community resilience was obtained. This provides for a more scientific understanding of how these factors affect community resilience in the context of COVID-19. This also supports the development of useful guidance toward improving the management of community resilience in the COVID-19 era.

## 2. Methodology

The interpretative structural model (ISM) allows the internal structure of a system to be revealed by processing known but messy system–element relationships [34] and was put forward in 1974 by Warfield [35]. Its basic principle is to decompose the constituent elements of a complex system into several sub-elements. Drawing on a combination of theoretical knowledge, practical experience, and statistical analysis, the system elements are made into a multi-level hierarchical structure diagram. The analytic hierarchy process (AHP) is an effective method for quantitative analysis that was put forward by American operational research scientist, Professor Saaty of Pittsburgh University, in the early 1977s [36,37]. This method emphasizes the mutual influence and restrictions among various factors.

In this study, firstly, an evaluation index of urban community resilience was developed, including the direct factors, indirect factors, and deep-seated fundamental factors affecting community resilience and found by using ISM. Then, based on the multi-level hierarchical structure obtained by ISM, the AHP structure chart was established. To establish the importance of the evaluation indexes that affect community resilience, the relative weight of each evaluation index was obtained by using Yaahp software (Shanxi yuan decision software technology Co., Ltd, Shanxi, China). The specific principles are shown in Figure 1.

### 2.1. ISM Method

Step 1: Establish adjacency matrix.

The factors found to influence community resilience are denoted as Y_1,_ Y_2,_ …Y_n_, *n* is the quantity of the resilience influencing factors, and Y is the set of resilience influencing factors. The directed graph *G* can be described as a mathematical formula [38]:(1)$$G=\left\{\left(Y,R\right)\left|Y=n,R=m\right.\right\},$$
where $Y=\left\{Y_{1},Y_{2},\cdots ,Y_{n}\right\}$ and $R=\left\{\left(Y_{i},Y_{j}\right)\left|Y_{i},Y_{j}\in Y\right.\right\}$, the directed graph model describes the interrelationship between the elements of the influencing factors. A directed graph model was created to construct the adjacency matrix and the reachable matrix.

The factors were compared using the Delphi method to attribute scores to represent the degree of influence of each factor and to establish the adjacency matrix. The relationship between two factors in a directed graph G can be represented by an n×n adjacency matrix $A=\left(a_{ij}\right)$
(2)$$a_{ij}=\left\{\begin{array}{c} 1,\;\left(Y_{i},Y_{j}\right)\in R\\ 0,\;\mathrm{other} \end{array}\right.,$$

Step 2: Calculate the reachability matrix B.

The reachable matrix B can be obtained by processing adjacency matrix A with Boolean operation rules.
(3)$$B\equiv {\left(A+I\right)}^{n+1}={\left(A+I\right)}^{n}\neq \cdots \neq {\left(A+I\right)}^{2}\neq \left(A+I\right),$$

The reachability matrix reflects the structural relationship between the influencing factors after continuous influence.

Step 3: Decomposition reachability matrix B.

By decomposing the reachable matrix to construct the structure analysis model, an hierarchical structure diagram was established.

### 2.2. Calculate the Relevant Weight (AHP) of all Levels of Influencing Factors

The AHP combines qualitative analysis with quantitative analysis and the determination of weights is obtained based on the relative importance of the various factors. The AHP method was then used to calculate the relative weight of each influencing factor. The specific steps are as follows:

Step 1: Establish the evaluation model of urban community resilience. A rational network is constructed by quantified correlations.

Step 2: On the basis of the evaluation model, based on Delphi method and Yaahp software,
(4)$$CR=\frac{CI}{RI}<0.1,$$

The judgment matrix is constructed to calculate the weight, and the consistency test is conducted according to 0.1, and then the relative weight of each index of urban community resilience is determined.

## 3. Factors Influencing Disaster Community Resilience

### 3.1. Study Area

Wuhan, the capital city of Hubei province in China, is a mega city in central China and had a population of 11 million before the COVID-19 outbreak. In December 2019, there was an outbreak of pneumonia of unknown cause in Wuhan, with an initial epidemiological link to the local Huanan Seafood Wholesale Market, where there was also the sale of live animals. On 12 February 2020, the World Health Organization named this new corona virus COVID-19. As communities are the first to be exposed to disasters, the Chinese government responded by taking rapid public health measures to protect the wider community. Three communities in the city of Wuhan, namely the Guandong community in Wuchang, the Baibuting community in Hankou, and the Wangjiawan community in Hanyang, were selected as study areas in this research (Figure 2).

In this study, a combination of a questionnaire survey and field observations were used to collect individual responses from members of the community groups. To determine the sample size for the questionnaire survey, a simplified formula was applied as follows [39],
(5)$$n=\frac{t^{2}p(1-p)N}{N\Delta p^{2}+t^{2}p(1-p)},$$
where *n* is the smallest sample size, *N* is total population in study area, *t* is probability degree. According to correlating table, *t* equals 1.96 in 97% probability degrees. *p* represents a rounded number, *p* = 0.5. △*p* is acceptable error, △*p* = 6%.

### 3.2. Factors Selection

Firstly, the factors influencing disaster community resilience were selected by using a fuzzy Delphi method. Through a combination of the results of a review of related literature [26,40], seventeen factors to COVID-19 were found to influence community resilience and these were selected on the basis of a common framework [26]. Then the dimensions and indicators of community resilience to COVID-19 were identified based on the common framework and fuzzy Delphi method.

The fuzzy Delphi method uses statistical analysis and fuzzy operations to transform the subjective opinions of experts into objective data. The fuzzy Delphi method was used to select the factors, and it comprehensively considers the uncertainty and fuzziness of the expert′s subjective consideration, as required by this research. The judgments of experts on seventeen factors identified from the literature using a common framework were produced using semantic variables in an online questionnaire survey. To minimize the differences caused by their subjective judgments, experts were asked to provide both a possible lowest and a possible highest value using a score from 0 to 10, rather than just a random number.

Secondly, the fuzzy Delphi method was used to consult experts. The purpose of the survey is to evaluate the importance, order, and grade of the factors in the system. Each evaluation factor includes two parts: (1) importance degree: this is to evaluate the level of importance of this factor to the upper evaluation dimension; (2) acceptable range: this is the acceptable range to evaluate the importance of this factor to the upper evaluation dimension, elicited with both maximum and minimum values. The above values are provided on a scale of between 0 and 10, the larger the value, the higher the importance.

Based on the survey, the most conservative cognitive values, the most optimistic cognitive values, and single values were selected (refer to Table 1). Then the minimum values (C_L_, O_L_), geometric values (C_M_, O_M_), and maximum values (C_U_, O_U_) of the most conservative cognitive values and most optimistic cognitive values were calculated, respectively. Average values were also calculated for the minimum values, Min, and geometric values, Mix. M-Z represents the verification value in the gray area. When it is a positive number, it means that the expert opinion tends to be consistent and the evaluation index has reached convergence; otherwise, when it is negative, it means that the expert opinion is inconsistent and the evaluation index has not reached convergence, so the evaluation index should be put to a second survey. G represents the importance value of expert consensus, and the higher the value, the higher the degree and importance of expert consensus.

A threshold was set in order to select the more important factors from the common framework system for community resilience. In order to find a reasonable threshold value, the geometric mean of the minimum value, maximum value, and single value of all the evaluation factors to be screened was calculated again. For example, for criterion layer X_1_ and the secondary indicator layer Y_1_, Y_2_, and Y_3._

After calculation, the threshold value of this survey was established as 6.2. Then, the evaluation system was constructed using the selected factors. These were drawn from the five dimensions of urban community resilience, these being social capital, economic capital, physical environment, demographic characteristics, and institutional factors, as shown in Table 2.

### 3.3. Data Collection

Wuhan is composed of three former individual cities of Wuchang, Hankou, and Hanyang and a questionnaire survey was used to collect data from the three types of communities: Guandong community in Wuchang, Baibuting community in Hankou, and Wangjiawan community in Hanyang. The Guandong community is a relatively new community located in Wuchang; the Baibuting community is an old community with a long history in Hankou; and the Wangjiawang community is representative of Hanyang.

The questionnaire used in this study was designed based on the related literature and included the comprehensive sociodemographic background of the respondents, (such as gender, age, education, occupation, income) and the primary and secondary indicators of resilience as extracted from the literature (see Appendix A). Community resilience factors, for example, social network relationship, health insurance, awareness of risk, level of income, and community functioning, were also included. For each respondent, the average factor score was calculated.

The questionnaire adopted a five-point variable scale to gauge the respondents’ views. Ultimately, 12,130 valid questionnaires, which included 3982, 4130, and 4018 valid questionnaires from the Guandong community, the Baibuting community, and the Wangjiawang community, respectively, were recovered in this survey. A reliability test was carried out on the basis of the questionnaire results and the Cronbach’s alpha was 0.80, indicating that the survey results were of scientific credibility.

Based on a combination of the findings from Table 2, an evaluation model of urban community disaster resilience to COVID-19 was established as shown in Figure 3.

### 3.4. Key-Factors Analysis

Using the ISM method, the questionnaire data were sorted to obtain the relationship data between the influencing factors, and the adjacency matrix A was obtained as shown in Appendix B.

According to the adjacency matrix, the relationship between the factors was obtained, and the reachability matrix B between indexes was obtained by Boolean operation, as shown in Appendix C.

The interpretative structure model diagram of the multi-level hierarchical structure is obtained by dividing the levels. The multi-level hierarchical structure, composed of direct factors in the surface layer, indirect factors in the middle level, and fundamental factors in the deep layer, was obtained by an interpretative structure model (ISM), as shown in Figure 4.

As seen in Figure 4, the hierarchical division diagram clearly shows the elements of urban community resilience and their interaction. The surface direct factors include member trust, health insurance, fixed asset value, construction environment, and infrastructure, which directly affect the resilience of urban communities. Indirect factors include community identity, employment, income level, education level, health level, risk awareness, leadership, and community autonomy. These all indirectly impact community resilience, and at the same time reflect the constraints on the surface influencing factors. The fundamental factors include social network relations, natural environment, vulnerable groups, and government management and investment, reflecting the root cause and essence of factors influencing community resilience to COVID-19.

Based on Yaahp software, the judgment matrix was constructed and calculated, and the consistency was checked. Based on the above analytic hierarchy process, the total weight of each index of the factors was calculated, as shown in Figure 5.

It can be seen from Figure 5 that among the factors influencing urban community resilience, the population characteristics are the main factors, with the weight of 0.3130, followed by economic capital accounting for 0.2189. Across all levels, the main influencing factors are found to be social network relationship, member trust, income level, construction environment, infrastructure, vulnerable groups, and risk awareness. The four factors of vulnerable groups, construction environment, income level, and member trust account for a large proportion, among which the vulnerable group accounts for the largest weight of 0.1853.

Combined with the hierarchical structure model in Figure 4 and the total comparing weight analysis of each index in Figure 5, it can be concluded that there are not only a large number of factors affecting the resilience of urban communities but also a complex relationship among them.

Figure 6 shows the weighting of the influencing factors on each dimension including a comparison across the dimensions. Combined with the hierarchical relationship of each index factor in the ISM model, we can see the direct factors that affect the resilience of urban communities. This suggests we should consider a wide range of issues in helping to develop community disaster resilience in the face of a pandemic. The key to this is to consider ways to protect the vulnerable members of the community. The built environment including the design and planning of infrastructure is also of importance in creating a robust physical environment. In addition, strengthening the trust between community members is conducive to community unity. Among the indirect factors, the income level directly reflects the economic level of the community. Generally speaking, income levels are found to positively affect the economic capital of the community and consequently lead to improvements in disaster resilience. Community education and publicity will help to cultivate risk awareness and make the community have normalized self-organization and self-adaptive ability. Among the fundamental factors, vulnerable groups are often ignored in disaster prevention and mitigation research. Research shows that sudden disasters will cause greater harm to vulnerable groups such as women, children, the elderly and the disabled [41]. Thus, disaster research should advocate community demand-oriented, targeted management of people of different ages, genders, and health status, so as to improve the overall resilience of the community in the management of COVID-19.

In addition, the social network relationship includes not only the relationship between the members of the community, but also the relationship with the government and the outside world, especially in the urban communities. The establishment of good internal and external contacts and interaction in the community will help the community to obtain help from the outside quickly and minimize the losses to the community in all aspects. Therefore, the construction of a flexible urban community requires community planners to take key and targeted management measures based on the overall understanding of the composition of the individual factors that influence community resilience as well as their combined influence on resilience within the community.

## 4. Conclusions

This study adopted a quantitative approach to investigate the factors found to influence community resilience to COVID-19 and proposed an integrated model of ISM-AHP. ISM was used to establish the hierarchical structure of community resilience to COVID-19, in order to consider the interactive network relationship among factors found to affect community resilience. The main influencing factors of community disaster resilience were determined, namely vulnerable groups, building environment, income level, and trust. The model can be used to effectively reflect the focus of efforts to improve urban community resilience. This approach toward community resilience assessment can be applied to group decision-making methods in the management of COVID-19 and can also be applied to identify the interdependence and relationships among key factors affecting community resilience. Some major conclusions can be drawn as follows:

(1)The integrated model of AHP and ISM can be used to analyze the relations among factors that influence community resilience to COVID-19. A three-level evaluation network was constructed by ISM. The surface direct factors included member trust, health insurance, fixed asset value, construction environment, and infrastructure, which directly affect the resilience of urban communities; indirect factors included community identity, employment, income level, education level, health level, risk awareness, leadership, and community autonomy; the fundamental factors included social network relations, natural environment, vulnerable groups, and government management and investment, reflecting the root and essence of issues that influence community resilience in the management of COVID-19.(2)The weightings of each factor were calculated by using AHP. The demographic characteristics of the population were found to be the main factors, followed by economic capital accounting for 0.2189. Social network relationship, trust, income level, built environment, infrastructure, vulnerable groups, and risk awareness were the main factors found to influence the resilience of the community to COVID-19.

In this study, the ISM method and AHP method were combined to comprehensively and systematically consider the mutual influence among the evaluation indexes of urban community resilience and the importance of each index. This was determined based on the causality among the influencing factors, which provides a more scientific and reasoned analysis for the development of community disaster resilience. At the same time, this also provides useful guidance toward the effective planning and design of urban construction and infrastructure.

The community is the first to be impacted and also the first basic line of defense in terms of disaster response. Further, the resilience of the community response plays an important role in reducing the impact of pandemics such as COVID-19. This study confirmed that different communities in the same area have different levels of disaster resilience in the whole process of disaster response. At present, the community still has a strong dependence on the higher government in, for example implementing COVID-19 prevention measures, early warning and emergency response. The management of the COVID-19 pandemic is a highly complex form of system engineering. In the trend of increasingly diversified and complex disasters, it is difficult to effectively reduce the negative impact of COVID-19 by relying solely on top–down administrative interventions. In order to reduce the negative impacts of COVID-19, we must empower the community to develop the capability and capacity to prevent and resist disasters, including their ability to recover quickly from such events. That is, by improving the resilience of the community, the expansion of pandemics such as COVID-19 can be prevented, and the evolution into a global problem can be avoided. When a large-scale pandemic does break out, the further refinement of community resilience capability can help the grass-roots decision-makers to formulate appropriate scientific and rapid response strategies and reduce the impact of COVID-19.

## Figures and Tables

**Figure 1 ijerph-18-00088-f001:**
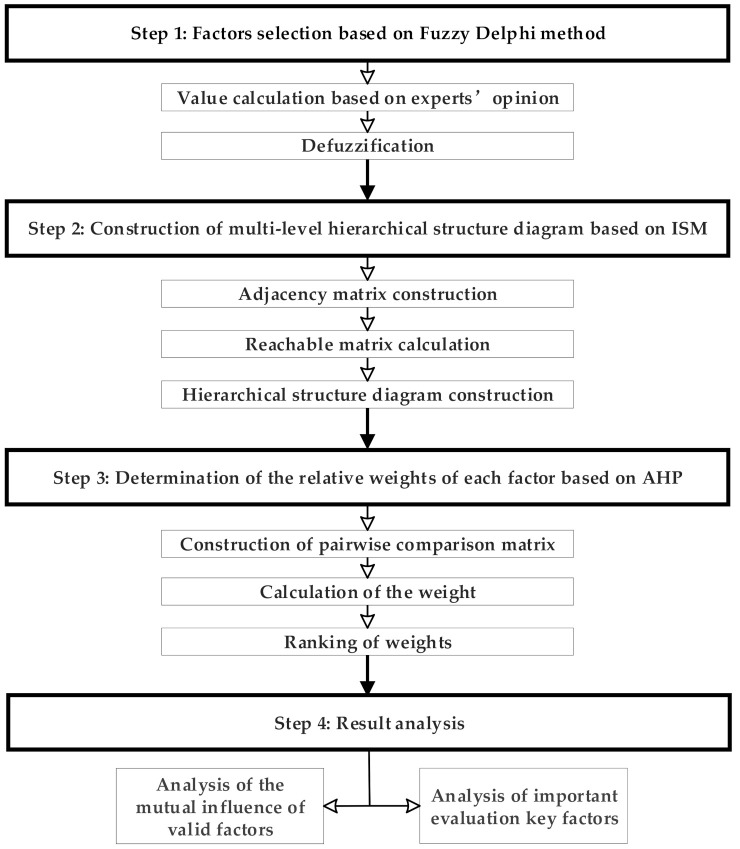
Factor analysis of urban community resilience based on interpretative structural model–analytic hierarchy process (ISM-AHP).

**Figure 2 ijerph-18-00088-f002:**
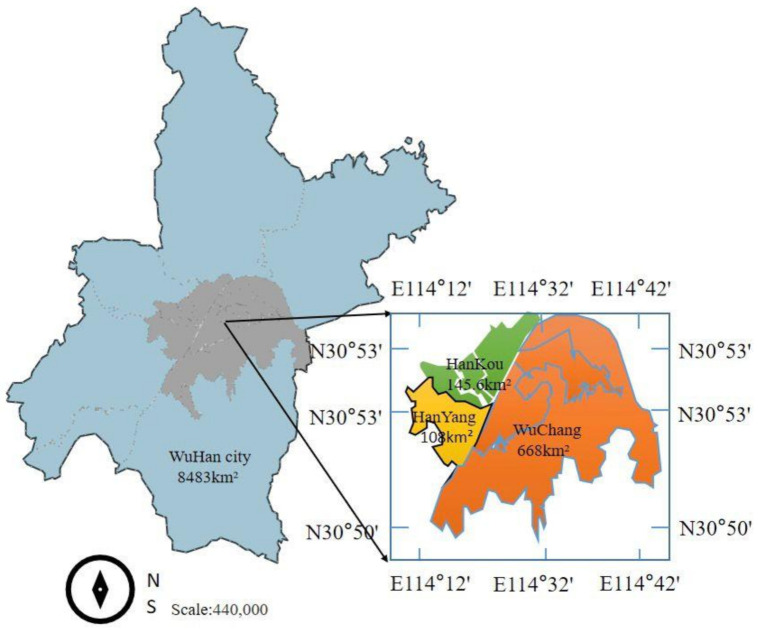
Map of the study area.

**Figure 3 ijerph-18-00088-f003:**
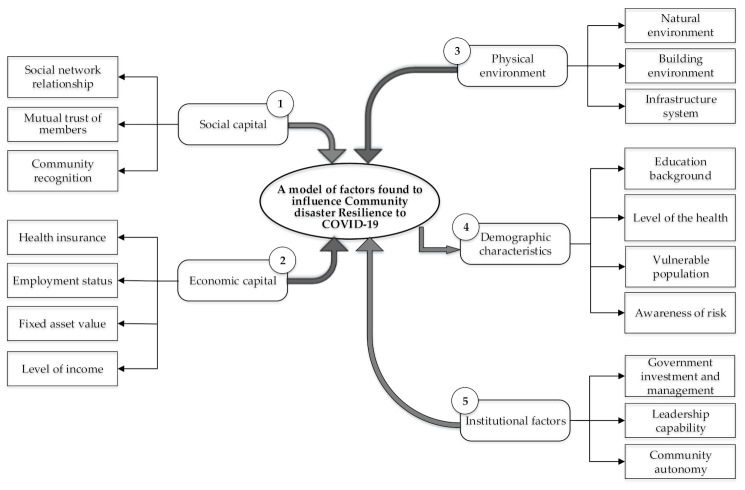
A model of factors found to influence community disaster resilience to COVID-19.

**Figure 4 ijerph-18-00088-f004:**
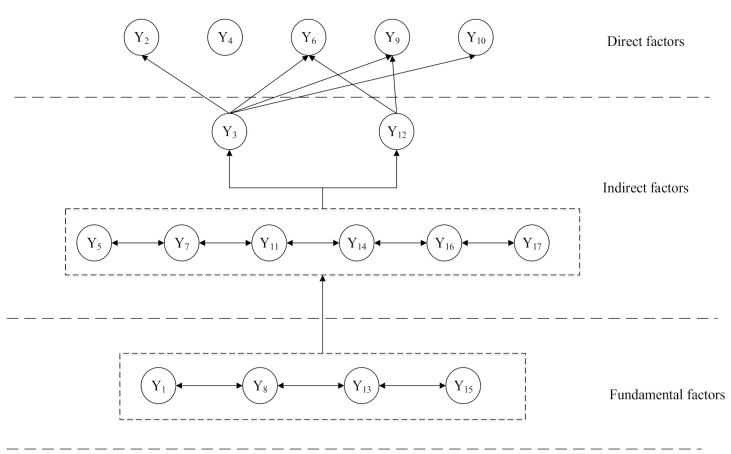
Multi-layer hierarchical diagram of influencing factors.

**Figure 5 ijerph-18-00088-f005:**
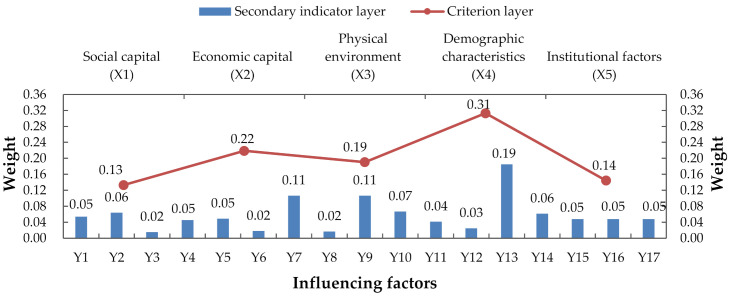
The total weight of factors of urban community resilience.

**Figure 6 ijerph-18-00088-f006:**
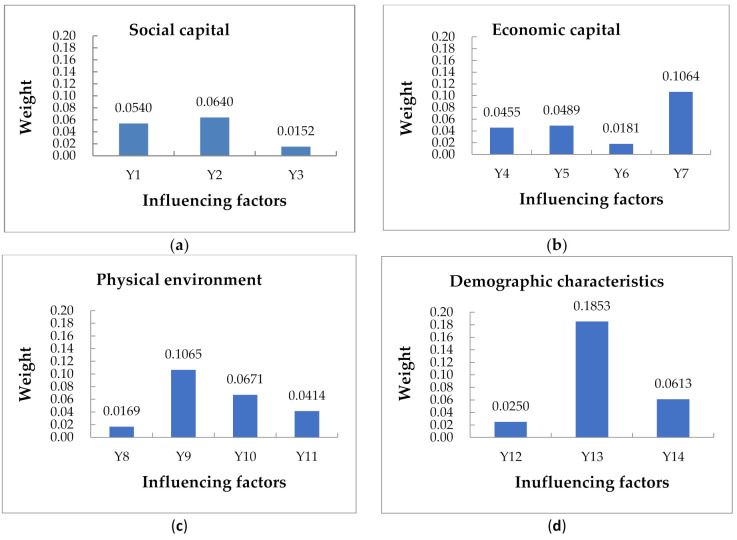
The weight of influencing factors of each dimension. (**a**) Social capital dimension; (**b**) economic capital dimension; (**c**) physical environment dimension; (**d**) demographic characteristics dimension; (**e**) institutional factors dimension; (**f**) comparison of each dimension.

**Table 1 ijerph-18-00088-t001:** Factor selection criteria.

Criterion Layer	Secondary Indicator Layer	C_L_	C_U_	O_L_	O_U_	Min	Mix	C_M_	O_M_	Average Value	M-Z	G	Selection
X_1_	Y_1_	1	8	5	10	3	10	3.87	7.81	6.24	0.94	6.21	√
	Y_2_	2	6	6	10	4	9	4.64	7.99	6.35	3.36	6.32	√
	Y_3_	1	8	5	10	4	10	4.01	7.89	6.51	0.89	6.26	√

C_L_, O_L_—minimum values; C_M_, O_M_—geometric values; C_U_, O_U_—maximum values; Min—average of minimum values; Mix—average of geometric values; M-Z—verification value in the gray area; G—importance value of expert opinion.

**Table 2 ijerph-18-00088-t002:** Multiple influencing factors of urban community resilience to COVID-19.

	Criterion Layer	Secondary Indicator Layer	Three-Level Indicator Layer
**Index system of influencing factors of community resilience**	Social capital (X_1_)	Social network relationship (Y_1_)	The relationship between the residents of the community (Y_11_)
			The relationship between residents and neighborhood committees (Y_12_)
			Community leader (Y_13_)
		Mutual trust of community members (Y_2_)	Trust among community members (Y_14_)
			Trust between members and government (Y_15_)
			Trust of members and social organizations (Y_16_)
			Trust between NGOs and the government (Y_17_)
		Community recognition (Y_3_)	Resident years of community members (Y_18_)
			Unique history and culture (Y_19_)
			Community members′ sense of belonging (Y_110_)
	Economic capital (X_2_)	Health insurance (Y_4_)	Medical insurance coverage (Y_21_)
		Employment status (Y_5_)	Employment rate (Y_22_)
			Diversification level of employment field (Y_23_)
		Fixed asset value (Y_6_)	Owner-occupied housing (Y_24_)
			Property value (Y_25_)
		Level of income (Y_7_)	Per capital income and household income (Y_26_)
			Diversification level of income sources (Y_27_)
			Income stability (Y_28_)
	Physical environment (X_3_)	Natural environment (Y_8_)	Location (Y_31_)
			The greening of the community (Y_32_)
			Sanitation status (Y_33_)
		Building environment (Y_9_)	Building density (Y_34_)
			Road condition (Y_35_)
			Community environment (Y_36_)
		Infrastructure system (Y_10_)	Emergency facilities (Y_37_)
			Fire station around the community (Y_38_)
	Demographic characteristics (X_4_)	Education background (Y_11_)	Population with high-school education (Y_41_)
			Population with professional and technical qualifications (Y_42_)
		Level of the health (Y_12_)	The physical and mental health of community residents (Y_43_)
		Vulnerable population (Y_13_)	Aging population (Y_44_)
			Disabled population (Y_45_)
			Transient population (Y_46_)
		Awareness of risk (Y_14_)	The risk awareness of community residents (Y_47_)
	Institutional factors (X_5_)	Government investment and management (Y_15_)	Manpower and material and financial support (Y_51_)
			Management mechanism (Y_52_)
		Leadership capability (Y_16_)	Neighborhood committee leadership (Y_53_)
		Community autonomy (Y_17_)	Social organizations (Y_54_)
			Community public participation (Y_55_)

NGOs: Non-Governmental Organizations.

## Data Availability

The data presented in this study are available on request from the corresponding author.

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
