# Peer review of "The Influence of COVID-19 on Community Disaster Resilience"

_ijerph, 2020, doi:10.3390/ijerph18010088_

Round 1
Reviewer 1 Report
Overall
- Linking community resilience to COVID-19 is a highly interesting and relevant topic. However, the paper does not (yet) clearly state how this is done methodologically. An assessment of community resilience is done, however without linking the factors to COVID, which is then only mentioned once in the discussion. The conclusions then try to link somehow COVID and the resilience assessment, but then also refer a lot to the methodology. Therefore the purpose of the paper remains blurry, also the conclusions are not fully convincing as they are based on an assessment which has not involved COVID in any form. While the paper topic is definitely interesting, my recommendation is that the authors need to emphasize the paper purpose stronger and then follow one approach, either a focus on the methodology and its suitability for such contexts, or on the assessment of community resilience for COVID.
- The link between an assessment of community resilience that doesn’t include any references to COVID and conclusions on the (measured?) influence of community resilience to COVID-19 remains extremely blurry. It is basically hypotheses, but not an outcome of the analysis? I recommend to add a few paragraphs on this, as it is the centerpiece of the paper
- I suggest to restructure the introduction as it sounds a bit strange, the first two sentences try to give a bigger picture, however they condensed so much information that it now links aspects which not necessarily together (see below)
- The introductory chapter explains well the use of resilience assessments in communities, however it doesn’t really explain the need to apply this for COVID – I suggest to add 1-2 sentences at the end of the chapter to state the demand/benefits of such approach
- There are quite some open questions around the analytical framework which needs to be specified more, e.g.: what kind of literature survey, how did you select the literature, given that the pandemic is an extremely recent event? Ow are the factors weighted, based on what? The theoretical basis isn’t explained, here another sub-chapter would be needed as the authors so far only focus on the calculation but not the factors chosen and their interlinkages/weighting. Another open question is how the five dimensions of urban community resilience (social capital, economic capital, physical environment, demographic characteristics and institutional factors) were selected, is this based on literature? The first two sound like you used the sustainable livelihood framework, but the latter 3 don’t. How were the indicators in table 1 put together, some of them don’t necessarily (directly) relate to COVID (e.g. fire stations), so it would be good elaborating a bit how/why they were chosen
- The case study would need some more introduction: how was it chosen, how were study participants selected, how was the questionnaires distributed
Specific
- 33f: please rephrase the introductory sentence, there has always been earthquakes, and even the other events you state – with the difference35f: that the hydro-met events and pandemics might get more frequent/severe
- 35f: this sounds a bit weird in relation to the before sentence – how would a health epidemic paralyse traffic?
- 44f: “The approaches adopted and experiences gained in China have provided useful learning for other countries to help with their prevention and response intervention.” Maybe skip that one as it’s a preliminary judgement (we are still in the midst of the pandemic) which is not absolutely needed for the paper
- 55f: the resilience concept presented here is a very technical one, I suggest to emphasize that it is more from this angle, e.g. used in infrastructure field – very much focusing on “bounce back” while recent publications (from social sciences) have a broader perspective
- 63: suggest not to use term “natural disaster” any more (https://www.nonaturaldisasters.com/)
- 82ff: this part is very interesting to read, have enjoyed it
- 106: ISM abbreviation needs to be introduced
- 121: why “a kind of” analytical model, sounds strange
- 188: recommend to have a new subchapter for the case study explanation
- Figure 2: scale missing
- 230ff: weighing process not fully clear
- General question on the survey: was it carried out before or during COVID? Needs to be added
Author Response
We greatly appreciate the excellent comments and suggestions from the referees and the Editorial Coordinator. We think that the paper has improved considerably with this revision.Please refer to the attachment for more details.

Reviewer 2 Report
From the literature, the authors identified and structured by importance the 17 potential factors that could explain community resilience after a major disaster such as the COVID-19 pandemic. Subsequently, the ISM with AHP was used to furtherly categorize the direct, indirect, and fundamental factors. This method demonstrated that the income level, population vulnerability, and the environment are the principal factors that drive community resilience under the impact of COVID-19.
These are my comments:
- Figure 2. The map of the area of study is missing coordinates, North, and scale. Also, the location of that area with respect to the globe should be included.
- Line 33: “In recent years, earthquakes, floods, droughts, public health epidemics and other emergencies have become frequent occurrences.” This statement needs to be supported by a reference.
- The numbering of the references in the Introduction section is not in order. Please review the order of the citations from 4 to 8.
- Line 71: “The Baseline Resilience Indicators for Communities (BRIC) model proposed by Cutter et al [17] is widely applied”. A few references of such applications should be included.
- Table 1 should be reformatted so that it stays in one full page.
- Line 199: “related evidence from the literature”. I am not sure that I understand what is the literature here mentioned. Was Figure 3 developed by the authors or it refers to previous studies?
- Tables 2 and 3 could be removed and replaced with a description of their content and meaning.
- Graphs in Figures 5 and 6 are missing axis labels. In Figure 5, I would replace the decimals to 2 so to make it more readable.
- Throughout the manuscript COVID 19 and COVID-19 appear. Please make it homogenous.
I will reconsider this manuscript for publication after major revisions.
Thank you.
Author Response
We greatly appreciate the excellent comments and suggestions from the referees and the Editorial Coordinator. We think that the paper has improved considerably with this revision. Please refer to the attachment for more details.

Round 2
Reviewer 2 Report
Thank you for making the revision as previously suggested. However, I still have a few comments:
1. From my previous revision, I noticed the following: Line 71: “The Baseline Resilience Indicators for Communities (BRIC) model proposed by Cutter et al [17] is widely applied”. A few references of such applications should be included. Authors Reply: "Thank you for your comments. We have added references in the revised version."
However, I still cannot see the references that support the statement: “is widely applied”. What are such applications? Please add a paragraph that reports a few examples of such applications of the widely used “Baseline Resilience Indicators for Communities (BRIC) model”.
2. Lines 123-125: “The Analytic Hierarchy Process (AHP) is an effective method for …” needs a citation to refer to the professor Satty’s work on the development of the AHP.
3. Figure 1 have some misspelled words, please revise.
4. The equations presented in section “2.1 ISM method” need references in the case that were not fully developed by the authors.
5. I would improve the representation of Figure 2 that shows the study area in the form of a map. As an example, the authors could refer to Figure 1 in “Belvederesi, C.; Dominic, J.A.; Hassan, Q.K.; Gupta, A.; Achari, G. Predicting River Flow Using an AI-Based Sequential Adaptive Neuro-Fuzzy Inference System. Water 2020, 12, 1622”, which shows the area of study with respect to the scale (in km), and the coordinate system for the whole area. I believe that this way there would be a better understanding of the size of the area of study, which is significantly large.
6. Figure 6 could be misleading: I would use the same numerical range for the vertical axes (for example from 0 to 0.20 for all the figures a-e to ease comparison among them. For example, as it is right now, it looks like Y9 and Y13 have similar impact, while Y13 is much larger than Y9. In other words, by replacing the current vertical axes, which differ from one graph to another at this time, with a single range from 0 to 0.20, it would help the reader to visualize the difference among such factors. Like in Figure 5, I would replace the decimals to 2 so to have a cleaner graphics.
7. In the conclusion section, I would add a paragraph to emphasize the contribution of your study, including what are the stakeholders that can benefit from your findings. In other words, who can benefit from this study and why.
Thank you for your valuable work.
Author Response
Thank you for your comments concerning our manuscript entitled. We are sending the revised m manuscript according to the comments of the reviewers. Revised portion are underlined in red.
